# Emerging Immunotherapy Approaches for Treating Prostate Cancer

**DOI:** 10.3390/ijms241814347

**Published:** 2023-09-20

**Authors:** Lingbin Meng, Yuanquan Yang, Amir Mortazavi, Jingsong Zhang

**Affiliations:** 1Division of Medical Oncology, Department of Internal Medicine, The Ohio State University Comprehensive Cancer Center, Columbus, OH 43210, USA; lingbin.meng@osumc.edu (L.M.); yuqanquan.yang@osumc.edu (Y.Y.); amir.mortazavi@osumc.edu (A.M.); 2Department of Genitourinary Oncology, H. Lee Moffitt Cancer Center & Research Institute, University of South Florida, 12902 USF Magnolia Drive, Tampa, FL 33612, USA

**Keywords:** immunotherapy, prostate cancer, immune checkpoint inhibitors, bispecific antibodies, CAR T cells, vaccines, cytokines, immunotherapeutic combinations

## Abstract

Immunotherapy has emerged as an important approach for cancer treatment, but its clinical efficacy has been limited in prostate cancer compared to other malignancies. This review summarizes key immunotherapy strategies under evaluation for prostate cancer, including immune checkpoint inhibitors, bispecific T cell-engaging antibodies, chimeric antigen receptor (CAR) T cells, therapeutic vaccines, and cytokines. For each modality, the rationale stemming from preclinical studies is discussed along with outcomes from completed clinical trials and strategies to improve clinical efficacy that are being tested in ongoing clinical trials. Imperative endeavors include biomarker discovery for patient selection, deciphering resistance mechanisms, refining cellular therapies such as CAR T cells, and early-stage intervention were reviewed. These ongoing efforts instill optimism that immunotherapy may eventually deliver significant clinical benefits and expand treatment options for patients with advanced prostate cancer.

## 1. Introduction

Prostate cancer is the second-most commonly diagnosed cancer and the fifth leading cause of cancer death globally [1,2]. In 2023, the United States is estimated to see about 268,490 new prostate cancer diagnoses and roughly 34,500 prostate cancer related deaths [1,2], with a 5 year survival rate surpassing 99% for localized cases [3]. Despite recent approval of life prolonging treatments such as sipuleucel-T, novel androgen receptor signaling inhibitors (ARSIs), cabazitaxel, Poly (ADP-ribose) polymerases (PARP) inhibitors, and radioligand therapies, more than two thirds of patients with metastatic prostate cancer died within 5 years [4,5]. There is an unmet need to develop new treatments for this lethal disease.

Unlike highly immunogenic tumors such as melanoma, prostate cancer poses unique challenges for immunotherapy [6,7]. The tumor microenvironment often displays immunosuppression, with enrichment of regulatory T cells (Tregs), myeloid-derived suppressor cells (MDSCs), and M2 macrophages, which can promote immune evasion [8]. Prostate tumors tend to harbor fewer neoantigens due to a relatively low tumor mutational burden, reducing their immunogenicity [9]. There is also marked upregulation of immunosuppressive molecules such as CTLA-4 and DcR3 that attenuates anti-tumor immunity [10]. Furthermore, TGFβ is widely expressed in prostate cancer tumor cells and stromal cells and the expression of TGFβ is enriched in stromal cells of castration resistant prostate cancer and bone metastases [11,12,13]. Consequently, prostate cancers are often resistant to immunotherapies such as checkpoint inhibitors that have shown efficacy in other malignancies. Comprehending these hurdles is vital for engineering more potent immunotherapies. Current research aims to gain deeper insights into the prostate cancer immunological landscape to create novel treatments that can overcome immunosuppression and trigger robust anti-tumor responses.

Since the approval of the first immunotherapy sipuleucel-T in treating metastatic castration-resistant prostate cancer in 2010, the past decade has witnessed a surge in immunotherapy trials in prostate cancer. The major immunotherapy approaches that have been tested encompass immune checkpoint inhibitors, bispecific T cell-engaging antibodies, adoptive cell therapy utilizing gene-modified T cells directed against cancer, and therapeutic vaccines priming tumor-specific immunity [14,15]. In this review, we will delve into the emerging immunotherapy approaches for prostate cancer, as illustrated in Figure 1. We will discuss the scientific rationale derived from preclinical studies, outcomes from early phase clinical trials, novel strategies under examination in ongoing trials, and potential future directions.

## 2. Immune Checkpoint Inhibitors

### 2.1. Background on CTLA-4, PD-1/PD-L1 Pathways

The advent of immune checkpoint inhibitors (ICIs) has dramatically transformed the landscape of oncology by enhancing intrinsic anti-neoplastic immunity. Two predominant immune checkpoint pathways that have been extensively studied in cancers are those associated with cytotoxic T-lymphocyte-associated protein 4 (CTLA-4) [16] and programmed cell death protein 1 (PD-1) [17,18]. CTLA-4 is a protein confined to the surface of T cells, where it supersedes the costimulatory receptor, cluster of differentiation (CD) 28 in binding to shared ligands CD80/CD86 present on antigen-presenting cells (APCs) [16]. This liaison propagates an inhibitory signal that represses T cell proliferation and the production of the cytokine Interleukin-2 (IL-2), a response that is typically facilitated by CD28-mediated costimulation [19]. Conversely, PD-1 is primarily manifested on activated T cells and interacts with its respective ligands PD-L1 and PD-L2, commonly upregulated on neoplastic cells or antigen presenting cells (APCs) [20,21]. The coupling of PD-1 with its ligands initiates inhibitory signals resulting in the diminished production of cytokines, cell proliferation, survival, and cytolytic activity of PD-1+ T cells within the tumor microenvironment [22,23].

In the setting of prostate cancer, upregulation of either PD-L1 or CTLA-4 may be indicative of a more aggressive disease course and an unfavorable prognosis [24,25]. Antibodies targeting immune checkpoints are contrived to enhance anti-tumor immunity by bypassing inhibitory pathways through the blockade of CTLA-4 or PD-1. Consequently, ICIs such as ipilimumab (anti-CTLA-4) [26] and anti-PD1 antibodies such as pembrolizumab and nivolumab (PD-1 inhibitors) [27] have been studied either as monotherapy or in combination to treat metastatic prostate cancer.

### 2.2. Key Clinical Trials of Immune Checkpoint Inhibitors in Prostate Cancer

Unlike the life prolonging activities seen in melanoma [28,29,30] and lung cancer [31,32], the clinical efficacy of ICIs for prostate cancer have been limited. In the Phase 1/2 CheckMate 063 trial (NCT00730639), nivolumab monotherapy demonstrated varying response rates among different cancers. Specifically, non-small cell lung cancer (NSCLC), melanoma, and renal cell carcinoma (RCC) patients had objective response rates of 18%, 28%, and 27%, respectively, as defined by RECIST criteria. However, there were no observed objective responses in those with colorectal or prostate cancer prostate cancer [6,7]. The Phase 1b KEYNOTE-028 trial examined pembrolizumab in PD-L1 positive advanced prostate cancer, reporting an objective response rate of 17.4% and disease control rate of 52.2% (NCT02054806) [33]. The Phase 2 KEYNOTE-199 trial exhibited low response rates of 3–5% with pembrolizumab in docetaxel-treated metastatic castration-resistant prostate cancer (mCRPC), although protracted responses surpassing 16 months were observed in some patients (NCT02787005) [34,35]. Exploratory biomarker analysis identified higher response rates in subsets with DNA damage repair (DDR) aberrations, including 11% in patients with BRCA1/2 or ATM mutations [35]. This suggests genomic alterations enabling higher tumor immunogenicity such as neoantigen burden and PD-L1 upregulation may enrich for ICI benefit. Overall, single-agent PD-1 blockade has deficient anti-tumor activity in advanced prostate cancer. Anti-CTLA4 with ipilimumab has been tested in two phase 3 trials in mCRPC. CA184-043 is conducted in the post docetaxel setting and involves 8 gray XRT to a bone lesion to induce an inflamed tumor microenvironment prior to treatment with ipilimumab [36]. Given chemotherapy could suppress the immune response, CA184-095 was subsequently conducted in pre docetaxel mCRPC. However, neither trial, when compared to a placebo, showed enhanced OS in mCRPC [37,38] (see Table 1).

Recognizing the lack of immune cell infiltration in “cold tumors” such as prostate cancer, a myriad of clinical trials have embarked on strategies of combining ICIs with chemotherapy, targeted therapy, and ARSI. Pre-clinical studies have shown that suppressing androgen signaling could remodel the prostate cancer microenvironment to enhance immune cell infiltration and delay CD8+ T cell exhaustion [39,40,41]. This strategy has been tested in the phase 3 trials combining atezolizumab/anti-PDL1 with AR antagonist, enzalutamide (IMbassador250/NCT030163120) or pembrolizumab/anti-PD1 with enzalutamide (KEYNOTE-641/NCT03834493) in mCRPC [38,39]. As shown in Table 1, neither study showed improvement in OS compared to enzalutamide alone. PARP inhibitors such as olaparib have been approved for treating mCRPC with deleterious mutations in DNA homologous recombination repair genes. Blocking DNA damage repair with PARPi could increase tumor mutation, neoantigen load, and enhance the efficacy of ICIs. The pembrolizumab and olaparib combination was therefore tested in Phase 3 Keynote-010. Of note, the trial was conducted in a biomarker unselected mCRPC patient population and these patients have progressed through docetaxel and one, but not two ARSI therapies. No improvement in rPFS or OS was noted in this study [42]. In solid tumors such as non-small cell lung cancer, adding chemotherapy to ICI has been shown to improve OS presumably through converting the “cold tumor” to “hot tumor” with chemotherapy. Docetaxel is the only frontline chemotherapy that is approved for metastatic prostate cancer based on improvement of OS. The docetaxel pembrolizumab combination was compared to docetaxel in the phase 3 Keynote-921 trial (NCT03834506) in mCRPC [43,44]. No improvement in OS was noted. In addition, recent press releases on the Phase 3 CheckMate -7DX trial (NCT04100018) report that adding nivolumab to docetaxel did not improve outcomes when compared to using docetaxel alone in patients with mCRPC. The trial did not meet the primary endpoints of radiographic progressive free survival (rPFS) or OS at its final analysis [45]. Collectively, these Phase 3 trials involving checkpoint inhibitors, either as monotherapies or in combination regimens, have not substantiated any survival improvement in mCRPC, as summarized in Table 1.

While most phase III trials have not shown a definitive survival advantage, preliminary signs of immunotherapy activity have been observed in subsets of mCRPC patients, particularly in earlier phase studies. For instance, in a comparative effectiveness analysis of 741 mCRPC patients, ICIs improved outcomes versus taxanes for those with tumor mutational burden (TMB) ≥10 mutations/megabase, indicating TMB may predict ICI benefit [46]. Additionally, another study highlights that Ipilimumab can be effective in treating patients with CRPC, with some showing significant responses and no traceable residual disease [47]. In another study, pembrolizumab yielded PSA50 declines in 53% of MSI-high mCRPC cases, although the small sample size warrants further research [48]. Furthermore, the COSMIC-021 phase 1b trial combining cabozantinib and atezolizumab showed encouraging activity, with an objective response rate of 23% in mCRPC. However, 55% experienced grade 3–4 adverse events, with pulmonary embolism, diarrhea, fatigue, and hypertension being the most prevalent [49].

The limited efficacy of immune checkpoint inhibitors in prostate cancer can be attributed to multifarious factors, including the relatively low tumor mutation burden of prostate cancers, the cold tumor microenvironment with attenuated CD8+ T cell infiltration and diminished Major Histocompatibility Complex (MHC) class I expression [46,50,51,52]. Prostate cancers have also been shown to secrete factors such as transforming growth factor beta (TGFβ), interleukin-10 (IL-10), and vascular endothelial growth Factor (VEGF) [53,54], constructing an immunosuppressive milieu that may demonstrate resistance to immunotherapy. Therefore, it is critical to perpetuate research efforts to uncover innovative strategies to augment the efficacy of these inhibitors in prostate cancer.

**Table 1 ijms-24-14347-t001:** Phase 3 clinical trials evaluating immune checkpoint inhibitors with unimproved survival outcomes in mCRPC patients.

NCT Number	Trial Name	Phase	Patients	Description	Results
NCT00861614	CA184-043	3	988	Ipilimumab + RT vs. placebo + RT in mCRPC	Median OS with ipilimumab was 11.2 months (95% CI 9.5–12.7) compared to 10.0 months (95% CI 8.3–11) on placebo. The HR was 0.85 (95% CI 0.72–1.00) with a *p*-value of 0.053 [36].
NCT01057810	CA184-095	3	837	Ipilimumab vs. placebo in mCRPC	Median OS with ipilimumab was 28.7 months (95% CI 24.5–32.5) compared to 29.7 months (95% CI 26.1–34.2) on placebo. The HR was 1.11 (95.87% CI 0.88–1.39) with a *p*-value of 0.3667 [55].
NCT03834493	KEYNOTE-641	3	1244	Pembrolizumab + enzalutamide vs. placebo + enzalutamide in mCRPC	Primary endpoints were not met [56].
NCT03834519	KEYNOTE-010	3	793	Pembrolizumab + olaparib vs. NHA in mCRPC	Median OS with Pembrolizumab + Olaparib was 15.8 months (95% CI 14.6–17.0) compared to 14.6 months (95% CI 12.6–17.3) in the control arm. The HR was 0.94 (95% CI 0.77–1.14) with a *p*-value of 0.26 [42].
NCT03834506	Keynote-921	3	1030	Pembrolizumab + docetaxel vs. docetaxel in mCRPC	Median OS with Pembrolizumab + Docetaxel was 19.6 months (95% CI: 18.2 to 20.9) compared to 19.0 months (95% CI: 17.9 to 20.9) with Docetaxel alone. The HR was 0.92 (95% CI 0.78–1.09) with a *p*-value of 0.1677 [57].
NCT03016312	IMbassador250	3	772	Atezolizumab + enzalutamide vs. placebo + enzalutamide in mCRPC	Median OS with atezolizumab + enzalutamide was 15.2 months (95% CI 14.0–17.0) compared to 16.6 months (95% CI 14.7–18.4) in the control group. The HR was 1.12 (95% CI 0.91–1.37) with a *p*-value of 0.28 [58].
NCT04100018	CheckMate -7DX	3	984	Nivolumab + docetaxel vs. Placebo + docetaxel in mCRPC	Primary endpoints were not met [59].

Abbreviations: NCT: National Clinical Trial; NHA: Next-generation Hormonal Agent Monotherapy; OS: Overall Survival; RT: Radiotherapy.

### 2.3. Ongoing Clinical Trials with Immune Checkpoint Inhibitors in Prostate Cancer

Despite initial challenges, several ongoing clinical trials are investigating the use of CTLA-4 or PD-1/PD-L1 inhibitors in castration-sensitive prostate cancer and low disease burden when the host is presumed to have a stronger immune system (NCT03007732). ICIs are also being tested in combination with radioligand therapy, which could elicit broader and deeper inflamed tumor microenvironments compared to targeted external beam radiation (NCT03093428, NCT05766371, NCT05150236). Strategies to enhance efficacy include the incorporation of radiotherapy (NCT03093428 and NCT05766371), chemotherapy (NCT04709276), and radical prostatectomy (NCT04565496). Table 2 provides an overview of these key ongoing clinical trials. These combined approaches aim to foster synergistic immune-mediated anti-tumor activity through complementary mechanisms and ameliorate the immunosuppressive influences endemic in prostate tumors. Future directions for enhancing the application of immune checkpoint inhibitors in prostate cancer entail the development of rationally tailored combination regimens specific to individual immune phenotypes or genomic contexts, defining mechanisms of primary and acquired resistance to inform innovative therapeutic strategies, and refining the sequencing of androgen deprivation or chemotherapy regimens.

Simultaneously, the quest to identify novel immune checkpoint targets beyond the paradigmatic CTLA-4 and PD-1/PD-L1 pathways is ongoing. Emerging targets such as lymphocyte activation gene 3 (LAG-3) [60,61], T-cell immunoglobulin and mucin-domain containing-3 (TIM-3) [62], and V-domain Ig suppressor of T cell activation (VISTA) [63] are being evaluated as potential inhibitors of anti-tumor immunity within the context of prostate tumors, either independently or in synergy with PD-1 signaling. While initial findings from monotherapy studies involving approved CTLA-4 and PD-1 inhibitors have not demonstrated substantial benefits in unselected mCRPC patients [64,65], continued investigation guided by the identification of suitable biomarkers, rationally designed combination strategies, sequence optimization, and their application in earlier stages of the disease may yet herald clinical benefits from immune checkpoint inhibitors in prostate cancer. Insights gleaned from past unsuccessful trials will help shape more effective clinical trial designs and immunotherapy strategies in the future. Despite existing challenges, the rapid advances in our understanding of the complex interplay between tumors and the immune system engender cautious optimism about the future prospects of immune checkpoint inhibitors in prostate cancer treatment.

## 3. Bispecific Antibodies Targeting T Cell Costimulatory Receptors

### 3.1. Background on Bispecific Antibodies

Bispecific antibodies denote an evolving and auspicious array of precision cancer immunotherapeutics, which possess the capacity to concurrently target two antigens or epitopes via a singular therapeutic entity [66,67]. These are meticulously engineered constructs that incorporate antigen-binding domains derived from two distinctive monoclonal antibodies. This dual-targeting capability facilitates the simultaneous engagement of a tumor cell, via a cancer-specific antigen, and a cytotoxic T cell, through a costimulatory receptor such as CD3, CD28, or CD137 [68,69]. The basic structure of a bispecific antibody is constituted by two divergent heavy chain–light chain pairs, each obtained from a monospecific antibody [67]. A variety of molecular formats have been conceived to bolster the bispecific binding, including tandem single-chain variable fragments (taFv) [70], dual-affinity retargeting (DART) molecules [71], and bi/tri-specific T cell-engagers (BiTEs/TriTEs) [72,73]. The meticulous optimization of antibody engineering is pivotal for achieving the desired target binding, stability, and pharmacokinetics.

The bispecific antibody’s dual targeting of a tumor antigen and a T cell costimulatory receptor bestows potent activation, proliferation, and cytotoxic function signals to T cells in an antigen-dependent manner [67,68,74]. Upon binding of the bispecific antibody to its designated tumor antigen, an immunological synapse is formed through the crosslinking of the cancer cell and T cell. Following this, the ligation of the costimulatory receptor such as CD28 or CD3 delivers a stimulatory signal, leading to the activation of the adjacent cytotoxic T cell [68,69]. This sequence of events instigates antigen-specific T cell immunity, which is directly targeted at malignant cells expressing the pertinent antigen.

### 3.2. BiTEs Underdevelopment in Prostate Cancer

An increasing assortment of bispecific antibody constructs, which link prostate cancer-associated antigens to T cell costimulatory receptors including CD3, CD28, and CD137, have undergone evaluation within preclinical models [69,75,76]. These pioneering immunotherapies have demonstrated potent antineoplastic effects in humanized mouse xenograft models as well as indepth mechanistic in vitro analyses. For instance, a PSMA × CD28 bispecific antibody facilitated polyclonal T cell proliferation, proinflammatory cytokine secretion such as IL-2 and interferon-gamma, and eradication of PSMA-expressing tumors in murine models, yielding a statistically significant extension in survival [77]. Similarly, PSMA × CD3 bispecific antibody constructs triggered Granzyme B and perforin-mediated prostate cancer cytolysis along with the release of cytokines such as IL-2 and interferon-gamma from redirected human T lymphocytes in co-culture assays [78]. These therapeutics also conferred dose-dependent tumor growth inhibition in vivo, albeit exhibiting diminished efficacy against malignancies demonstrating lower antigen expression levels.

AMG160, a bispecific T-cell engager (BiTE) antibody targeting the PSMA and CD3, boasts an extended half-life in comparison to first-generation BiTEs [79]. Animal model-based preclinical studies have revealed that AMG160 can induce T-cell mediated killing of prostate cancer cells expressing PSMA [79,80]. According to a phase 1 clinical trial (NCT03792841) reported at the ESMO meeting 2020 [81], dose dependent PSA decline was observed in 24/35 evaluable patients and 12/35 (34.3%) had a PSA reduction of more than 50%. Furthermore, 23.1% of patients experienced the disappearance of previously detectable circulating tumor cells during the treatment, indicating a favorable effect on disease burden. 30/43 patients experienced cytokine release syndrome (CRS), which was manageable and was most severe in cycle 1. 11/43 (25.6%) had grade 3 CRS as the worst grade. None of the treatment related adverse events led to treatment discontinuation. More recently, Amgen is replacing AMG160 with AMG340, another anti-PSMA × CD3 BiTE for clinical development in mCRPC. The phase 1 multi-center study on AMG340 is still ongoing. Other PSMA targeting BiTEs under early phase clinical development include Regeneron’s anti-PSMA × CD28 plus cemiplimab (anti-PD1) and LAVA therapeutics’ anti-PSMA gamma-delta T cell engagers, LAVA-1207 (NCT05369000).

Besides PSMA, novel BiTE constructs are under development exploring alternative tumor antigens such as Glypican-1, disintegrin and metalloproteinase 17 (ADAM17) [82,83]. Additionally, the prostate stem cell antigen (PSCA), overexpressed in a range of malignancies including mCRPC, is the target of GEM3PSCA, an affinity-tailored T cell adaptor that is currently being evaluated in a phase 1 clinical trial (NCT03927573) for PSCA-positive cancer [84]. However, this trial was terminated by the sponsor. JNJ-78278343 is a BiTE that targets human Kallikrein 2 (KLK2) and CD3. Notably, JNJ-78278343 can be delivered through subcutaneous injection, subcutaneous (SC) infusion or intravenous (IV) infusion. It is currently under development for metastatic prostate cancer (NCT04898634). Six-transmembrane epithelial antigen of prostate 1 (STEAP1) is another antigen that is being targeted in an ongoing phase 1 study with AMG509/anti-STEAP1xCD3 in mCRPC (NCT04221542). Both SC and IV delivery of AMG509 will be evaluated in this trial. In part 4 of this study, AMG509 will be tested in combination with either abiraterone or enzalutamide as a frontline option for mCRPC [85].

In tandem with ongoing research, endeavors are also in place to address the issue of the limited half-life of certain therapies. A proposed approach involves the utilization of an injectable polymer depot that releases anti-PSMA-BiTEs as the biopolymer gradually degrades. Preclinical trials employing mouse xenograft models of prostate cancer have demonstrated that this technique effectively sustains BiTE plasma concentration and enhances tumor growth inhibition, especially in tumors exhibiting low PSMA expression [80,86]. Notably, tarlatamab/AMG 757, a half-life extended BiTE therapy targeting delta-like ligand 3 (DLL3), has exhibited potential in preclinical models [87,88]. Considering the upregulation of DLL3 in neuroendocrine prostate cancer, AMG 757 represents a promising therapeutic option specifically for this form of cancer [87]. A phase 1 clinical trial (NCT04702737) is currently assessing AMG 757 in patients with de novo or treatment emergent neuroendocrine prostate cancer, with a focus on determining its efficacy and safety profile [89,90].

BiTE therapies do carry certain pitfalls, including a short half-life that requires frequent dosing, the risk of cytokine release syndrome that often results in patient hospitalization during cycle 1, and the lack of durable responses due to the development of resistance [91,92,93]. CC-1 was one of the early anti-PSMA × CD3 BiTEs in clinical development. During the phase 1 study of CC-1 (NCT04104607), prophylaxis with tocilizumab, an interleukin-6 receptor (IL-6) blocker, is required [94]. Although no grade 3 or above CRS was noted, CRS was observed in 79% of the 14 treated patients despite prophylaxis with tocilizumab [95]. Of note, high grade CRS is much less common with BiTEs compared to CAR-T therapies. Most ongoing studies have now foregone the requirements for prophylaxis with anti-IL6. Given that most severe CRS tend to occur during cycle 1, most studies require overnight observation during the first cycle of BiTE infusions. Table 3 summarized some representative ongoing BiTE trials in metastatic prostate cancer, and future investigations should continue to refine optimal designs, dosing, and combination strategies to propel the most promising candidates toward clinical translation.

## 4. Chimeric Antigen Receptor (CAR) T Cell Therapy

### 4.1. Background on CAR-T Cell Approach

CAR engineered T cell therapy has emerged as a promising form of adoptive cell immunotherapy wherein patient-derived T lymphocytes are genetically modified to express synthetic receptors directed against tumor-associated antigens (TAAs) [96]. These CAR constructs comprise an extracellular target binding domain originating from a tumor-specific monoclonal antibody single chain variable fragment (scFv), connected to intracellular T cell signaling domains capable of activating T cell effector functions [96,97]. Patient autologous T cells are isolated via leukapheresis and then genetically transduced to express the CAR using viral vectors ex vivo. The modified CAR T cells undergo expansion culture and are subsequently reinfused into the patient, where they can recognize and potently eliminate antigen-expressing cancer cells in an MHC-independent manner [97].

Second-generation CARs contain an intracellular costimulatory domain in addition to the CD3ζ signaling domain found in first-generation CARs [98,99]. This costimulatory domain enhances the activation and proliferation of the CAR T cells, leading to improved antitumor activity. Third-generation CARs contain two intracellular costimulatory domains, which can further enhance the activation and proliferation of the CAR T cells [100,101]. These costimulatory domains can be derived from various proteins, such as CD28, CD134, or CD137 [101]. Preclinical studies have shown that third-generation CARs can lead to improved antitumor activity compared to second-generation CARs [102].

Solid tumors pose unique challenges to treatment with CAR-T cells, including high antigen heterogeneity and the ability of tumors to escape from CAR-T cells [103]. Despite these challenges, researchers are making progress in developing CAR T-cell therapies for solid tumors by identifying tumor-specific proteins that can be targeted without harming healthy organs [103]. For prostate cancer, CARs have been engineered to target antigens overexpressed on prostate tumors compared to normal tissue, including PSMA [104,105], PSCA [106,107] and epithelial cell adhesion molecule (EpCAM) [108].

### 4.2. Preclinical Studies of CAR-T in Prostate Cancer

Serval pre-clinical studies with in vitro and in vivo models have shown the promise of CAR-T therapy in prostate cancer [109,110]. PSMA is the most-studied CAR-T target for metastatic prostate cancer. To overcome the immunosuppressive microenvironment of prostate cancer, PSMA targeting CAR-T cells that co-express dominant negative transforming growth factor β (TGF-β) type II receptor have been developed, and enhanced tumor eradication were seen in prostate cancer mouse models [111,112]. Prostate stem-cell antigen (PSCA) is another tumor antigen that is overexpressed in prostate cancer. An earlier preclinical study by Priceman et al. reported replacing the CD28 costimulatory signaling domain with 4-1BB of the PSCA-CAR enhanced T cell persistence and antitumor activity in a patient-derived bone metastatic mouse xenograft model [113]. Unlike the conventional strategy of using αβ type T cell receptors for CAR-T expression, Frieling et al. recently reported the feasibility and superiority of using γδ T cells. The ability of γδ T cells to recognize the phosphoantigens accumulated in the microenvironment of bone metastases also led to synergy with zoledronic acid [110]. Six-Transmembrane Epithelial Antigen of the Prostate 1 (STEAP1) is a surface antigen that is expressed in over 80% of metastatic prostate cancer, higher than the 60% positivity on an immunochemistry study conducted at Fred Hutchinson Cancer Center using a H score of 30 as a cutoff for positivity. The adoptive transfer of STEAP1 CAR T cells was associated with prolonged peripheral persistence and either disease eradication or substantial tumor growth inhibition in mouse metastatic prostate cancer models. Loss of the STEAP1 antigen was associated with recurrent or progressive disease [114]. This preclinical study also evaluated adding tumor-localized interleukin-12 (IL-12) therapy in the form of a collagen-binding domain (CBD)-IL-12 fusion protein to enhance the antitumor efficacy of the STEAP1 CAR T cell therapy [114,115].

Other strategies to improve the efficacy of CAR-T therapy in metastatic prostate cancer include developing dual-antigen CARs to mitigate antigen escape and tumor heterogeneity [116]. Moreover, “armored” CARs, co-expressing cytokines such as IL-12, IL-15, or IL-7/Chemokine (C-C motif) ligand 19 (CCL19), have demonstrated superior proliferative capacity, metabolic fitness, persistence, and anti-tumor efficacy compared to conventional CARs devoid of cytokine support [117,118]. The strategic combination of prostate cancer-targeted CAR T cells with PD-1 checkpoint inhibition or immunomodulatory drugs has also been studied in pre-clinical models [119].

The comprehensive optimization of processes surrounding CAR T cell manufacturing, lymphodepletion regimens, costimulatory domain selection, cytokine engineering, and combinatorial approaches is actively pursued to enhance the functional potency of the final CAR T cell products [120,121,122]. The deployment of toxicity models using humanized mice is imperative to delineate dosing strategies that limit on-target/off-tumor adverse effects against normal prostate tissue expressing shared antigens [123]. In summation, rigorous preclinical studies collectively underscore the promising efficacy of CAR T cell immunotherapy as a precision medicine approach for prostate cancer. These studies also illuminate key areas for optimization concerning CAR construct design, combinatorial strategies, and dosing, all of which are integral to maximizing therapeutic benefit.

### 4.3. Ongoing Clinical Trials of CAR-T Cell Therapy in Prostate Cancer

Several phase I studies have studied the safety and preliminary efficacy of CAR-T-based therapy for metastatic castration resistant prostate cancer (Table 4). The PSMA emerges as a propitious target due to its consistent membrane expression across a significant majority of mCRPC cells [114]. However, its expression in other tissues such as the small intestine, kidney, and salivary glands necessitates vigilance regarding potential on-target off-tumor toxicity in the context of PSMA-directed CAR-T therapy [124]. Other targets of interest include the PSCA (NCT05732948 and NCT05805371) and kallikrein 2 (KLK2) (NCT05022849), both of which display high expression in prostate cancer (Table 4) [125,126,127,128,129]. Current investigative endeavors seek to refine CAR design, specificity, and signaling to achieve potent antitumor efficacy with mitigated toxicities. The evaluation of efficacy and safety of CAR T cell therapy in prostate cancer remains a key component of ongoing clinical translation efforts.

Table 4 encapsulates representative phase 1 clinical trials investigating CAR-T cell therapy in advanced prostate cancer. Started in 2008, NCT00664196 is one of the earliest CAR-T trials in mCRPC. This study used non-myeloablative conditioning chemotherapy regimen prior to anti-PSMA CAR-T infusion, which is then followed by either low dose or high dose IL-2 IV infusion for a month [130]. Most observed toxicities were attributed to chemotherapy or IL-2 treatment. Notably, two out of five patients exhibited a PSA response. This study was suspended prior to completing accrual due to a lack of funding. NCT03089203 is a Phase 1 study evaluated the anti-PSMA CAR-T cells equipped with a dominant negative TGF-β receptor (CART-PSMA-TGFβRDN) [131]. The CAR-T starting dose for cohort 1 of this phase 1 trial is 1–3 × 10^7^ without lymph depletion chemotherapy, followed by cohort 2 at 1–3 × 10^8^. Lymphodepletion chemotherapy with cyclophosphamide at 300 mg/m^2^/day and fludarabine at 30 mg/m^2^/day will be added to 1–3 × 10^7^ CAR-T cells at cohort 3 and cohort 4 with 1–3 × 10^8^ cells. Both drugs will be given by IV infusion over 3 days at days -5, -4, and -3 prior to CAR-T infusion at day 0. This conditioning chemo regimen may reduce or eradicate tumor-infiltrating regulatory T cells, thereby impairing the ability of these suppressive T cells to inhibit adoptively transferred CAR-modified T cells [132,133]. Five of the 13 enrolled patients developed grade 2 or above cytokine release syndrome (CRS). One patient had >98% PSA reduction, but died from grade 4 CRS concurrent with sepsis. Three other patients had a PSA decline of more than 30% [131]. CRS indicates the resilience of CAR-T cells against the immunosuppressive characteristics of the tumor microenvironment and their ability to proliferate upon successful binding to the PSMA antigen. The importance of lymphodepletion chemotherapy was highlighted in another phase 1 study conducted at City of Hope, which evaluated PSCA-targeted 4-1BB-co-stimulated CAR-T therapy in mCRPC (NCT03873805). This study started with 1 × 10^8^ CAR-T cells without lymphodepletion chemotherapy, and no DLT or response was observed in the three-patient cohort. No DLTs and three stable diseases by RECIST criteria were observed in the cohort that added lymphodepletion chemotherapy with cyclophosphamide at 300 mg/m^2^/day. In the lymphodepletion chemotherapy cohort that included a higher dose of cyclophosphamide, two out of six patients developed a DLT of grade 3 cystitis. Four out of six patients in this cohort developed stable disease [106]. Moreover, NCT04227275 is a phase 1 study built upon NCT03089203 to further evaluate the safety and preliminary efficacy of the CART-PSMA-TGFβRDN therapy in mCRPC [134,135]. This study was stopped after enrolling nine patients when one patient developed grade 5 events of immune-effector cell-associated neurotoxicity syndrome and multiorgan failure after receiving 0.9 × 10^8^ CAR-T. Another patient in the cohort with 0.7 × 10^8^ CAR-T plus prophylactic anakinra, an IL-1 receptor antagonist, also experienced a grade 5 event likely related to immune toxicity [135].

Potential risks encompass CAR-T therapy allergic reactions during infusion, imbalances in blood mineral levels, heightened susceptibility to severe infections due to immunosuppression, cytopenias, Immune Effector Cell-Associated Neurotoxicity Syndrome (ICANS), and CRS [130,136]. CRS is a possible severe toxicity seen with immune therapies, particularly those engaging T-cells such as CAR-T therapy. The severity of CRS can vary and is graded based on criteria such as fever intensity, hypotension, and hypoxia according to the ASTCT consensus grading [137]. Symptoms of CRS may include fever, nausea, headache, tachycardia, hypotension, and respiratory distress [130]. ICANS represents another potential risk, manifesting as a clinical and neuropsychiatric syndrome post-immunotherapy administration. It is most commonly linked to immune effector cell (IEC) and T-cell engaging therapies [138]. The severity of ICANS can also be categorized, with grading being determined by parameters such as the Immune Effector Cell-Associated Encephalopathy (ICE) score, Cornell Assessment of Pediatric Delirium score, depressed consciousness levels, seizure occurrences, motor findings, and elevated intracranial pressure or cerebral edema [139]. CRS and ICANs have been pinpointed as the principal reasons for patient fatalities during early-phase CAR-T trials. Additionally, the efficacy of CAR T-cell therapy may not be universal as it depends on the accurate targeting of specific antigens expressed on cancer cells [140]. If the malignant cells lack the targeted antigens or possess mechanisms to evade immune recognition, the therapeutic efficacy may be undermined.

In conclusion, despite the promise shown by CAR T-cell therapy in preclinical studies and early clinical trials for prostate cancer treatment, further research is imperative to establish feasibility, safety, and potential efficacy signals for CAR T cell therapy approaches targeting different prostate cancer antigens. Further refinement of CAR design and manufacturing, combinational strategies, and predictive biomarkers will be critical to amplify clinical benefit.

## 5. Other Immune-Based Therapies

Beyond checkpoint inhibitors, bispecific antibodies, and CAR T cell therapy, an array of other immunotherapeutic strategies is under investigation for potential use in prostate cancer treatment. Crucial strategies encompass therapeutic vaccines, cytokines, and rational immunotherapeutic combinations.

### 5.1. Vaccines

Cancer vaccines represent a precise therapeutic approach designed to elicit robust tumor-specific CD4+ and CD8+ T cell responses against antigens associated with prostate cancer [141]. Table 5 summarized some key clinical trials of vaccines in the treatment of prostate cancer. One such vaccine is Sipuleucel-T (Provenge), an autologous dendritic cell vaccine amalgamated with a fusion protein that bridges prostatic acid phosphatase (PAP) to granulocyte-macrophage colony-stimulating factor (GM-CSF). The Phase 3 IMPACT trial (NCT00065442) demonstrated that Sipuleucel-T improved OS in patients with asymptomatic or minimally symptomatic mCRPC compared to a placebo, ultimately leading to FDA approval in 2010 [142]. Although only 5% of mCRPC patients had PSA response, the risk of death was reduced by 22.5% in this pivotal phase 3 trial with crossover design. This survival benefits were highest in patients with the lowest baseline PSA levels in the IMPACT trial as well the later PROCEED study [143]. Biomarker studies have shown enhanced peripheral immune response and antigen spread correlated with prolonged overall survival [144,145]. Sipuleucel-T treatment is well tolerated, with transient fever, back and joint pain, chills, fatigue, nausea, and headache as the common side effects.

Beyond dendritic cell vaccines, diverse other modalities are under exploration, including peptide [146], viral [147], bacterial [148], and nucleic acid-based (DNA or RNA) vaccines [149,150]. These tactics aim to catalyze immune recognition and the subsequent eradication of cancer cells. Peptide vaccines, which utilize petite protein fragments derived from TAAs or tumor-specific antigens (TSAs), exemplify this approach. These fragments can incite protective immunity against infectious or non-infectious diseases and serve as therapeutic cancer vaccines by invoking potent anti-tumor T-cell responses. Numerous peptide vaccines targeting prostate cancer antigens have embarked on early-phase clinical trials, with some rendering promising preliminary outcomes [151]. A remarkable example includes a vaccine targeting the Ras homolog gene family member C (RhoC), which has been found to generate a sustained T-cell immune response in a significant cohort of patients post radical prostatectomy (NCT03199872) [152]. Such advancements underscore the continued progress and potential of immunotherapeutic strategies in prostate cancer treatment.

Viral vector vaccines utilize a genetically modified virus to transport genetic material from the cancer cells into the body. This genetic material instructs the body’s immune system to identify and assail the cancer cells. In a clinical trial (NCT01322490) [123], a randomized, double-blind, phase 3 approach was used to evaluate the effectiveness of PROSTVAC-V/F +/− GM-CSF in individuals with asymptomatic or minimally symptomatic mCRPC [153]. The objective of this study is to ascertain whether PROSTVAC, either standalone or in conjunction with GM-CSF, is effective in prolonging OS in mCRPC patients. However, the trial did not meet its primary endpoint of improving OS compared to the placebo group [154].

DNA and RNA vaccines represent an innovative therapeutic approach that exploits genetic material derived from cancer cells to incite an immune response. These vaccines can be either administered directly or personalized by extracting immune cells from the patient, modifying them with specific genetic material, and reinfusing them [149,150]. In prostate cancer, the majority of DNA vaccines have targeted specific antigens such as PAP, PSA, or androgen receptor. The pursuit of other tumor-specific mutation-associated neoantigens has proven challenging, owing to the relatively low tumor mutational burden (TMB) characteristic of prostate cancer. A representative example of these endeavors is a plasmid DNA vaccine encoding human PAP (pTVG-HP), which has been investigated in several phase 1/2 trials (NCT04090528 and NCT03600350 in Table 6). However, to sustain an effective immune response, repeated vaccinations were required, and despite these efforts, tangible benefits were observed in a limited number of patients [149,150]. A subsequent phase 2 clinical trial focusing on non-metastatic high-risk prostate cancer with biochemical recurrence revealed improved 2 year metastasis-free survival only in a specific subgroup with rapid PSA doubling time (NCT01341652) [155]. This outcome highlights the importance of identifying patient subsets more likely to respond to DNA vaccine interventions and emphasizes the ongoing need for refined research and tailored strategies to maximize the potential benefits of DNA vaccines in prostate cancer therapy.

In summary, these different types of vaccines represent promising new approaches for the treatment of prostate cancer. Challenges to overcome include breaking immune tolerance and eliciting robust cytotoxic CD8+ T cell responses against self-antigens. More phase 3 trials are required to validate vaccines as effective standard treatments for prostate cancer in the future.

### 5.2. Cytokines

Cytokines, small proteins that are essential for immune system function, have been characterized as possessing anti-tumor properties in prostate cancer, acting as pivotal mediators in immune response and anti-tumor defense mechanisms [156]. However, their role in prostate cancer immunotherapy can exhibit a paradoxical nature. While certain cytokines are fundamental in establishing a defense against tumors, others may inadvertently facilitate tumor growth and inhibit anti-tumor responses [157]. This dual functionality underlines the complexity of their role in cancer treatment. The therapeutic efficiency of cytokine-based immunotherapy may be subject to modulation by factors such as the dosage administered, with the potential for unwelcome side effects at elevated doses, thereby further influencing treatment outcomes [158].

In addition, cytokines have been found to play a significant role in determining the effectiveness of other immunotherapeutic strategies in prostate cancer. For instance, one study illuminated the potency of peptide-sensitized dendritic cell-cytokine-induced killer (DC-CIK) cell preparations [159]. These experimental preparations demonstrated notable in vitro and in vivo antitumor activity against prostate cancer stem cell (PCSC)-enriched prostatospheroids [159]. Therefore, cytokines function as multifaceted agents within the immune system, playing intricate roles in both the promotion and suppression of anti-tumor responses. Their nuanced influence on prostate cancer treatment calls for additional investigation and meticulous consideration in therapeutic design.

Regarding the use of cytokines in prostate cancer treatment, interferon-alpha is a cytokine that has been demonstrated to have antiproliferative effects against prostate cancer cells in vitro [160]. The combination of cis-retinoic acid and interferon-alpha has been explored as a potential treatment for prostate cancer [161]. In vitro studies have shown that interferon-alpha, with or without cis-retinoic acid, has antiproliferative effects against prostate cancer cell lines: prostate adenocarcinoma cell line 3 (PC3) and Daudi lymphoblastoid cell line-145 (D-145) [160]. Androgen-independent prostate cancer cells often overexpress the anti-apoptotic protein B-cell lymphoma 2 (BCL2) [162], while retinoids can induce apoptosis and influence growth factor-beta in prostate cancer models [163]. By affecting proliferation, apoptosis, and cytokine signaling, these agents may hold therapeutic potential, although further research is required to translate these in vitro findings into clinical applications. In a word, cytokines have demonstrated promising anti-tumor effects in prostate cancer, but additional research is needed to thoroughly comprehend their potential benefits and limitations.

### 5.3. Immunotherapeutic Combinations

Immunotherapeutic combinations are based on the premise that their complementary and synergistic modes of action could potentially overcome the shortcomings of using a single agent [164]. This approach includes using checkpoint inhibitors alongside CAR T cells, bispecific antibodies, or vaccines to bolster T cell responses [165,166,167]. Chemo-immunotherapy strategies are also under exploration [168]. Identifying the optimal sequences and predictive biomarkers are currently key areas of active research [169]. Below are some key approaches on combination immunotherapies being explored in prostate cancer:Checkpoint inhibitors plus CAR T cells: CAR T cell function can be hampered by immunosuppressive factors in the tumor microenvironment [170]. Adding PD-1/PD-L1 checkpoint blockade aims to augment CAR T cell activation, proliferation, and persistence. Preliminary studies lend support to the notion that this approach can enhance efficacy [171].Checkpoint inhibitors plus bispecific antibodies: Bispecific antibodies also depend on T cell recruitment and activation. Combining with checkpoint blockade may boost the expansion and cytotoxic activity of bispecific antibody-redirected T cells against tumor cells [166,172].Checkpoint inhibitors plus vaccines: Vaccines prime tumor-specific T cells, while checkpoint inhibitors amplify their function [173]. Combined sequential administration is being explored clinically [174].Chemo-immunotherapy: Certain chemotherapies may stimulate immune pathways such as immunogenic cell death to complement immunotherapy effects. Docetaxel, cabazitaxel, and cisplatin combinations are being studied [175,176,177].

Several contemporary clinical trials are actively exploring the utilization of immunotherapy in the context of prostate cancer treatment, as delineated in Table 6. For instance, a phase 1 trial (NCT03532217) is engaged in assessing PROSTVAC in conjunction with checkpoint inhibitors ipilimumab/nivolumab and a neoantigen DNA vaccine for patients with metastatic hormone-naive prostate cancer [178]. Concurrently, a phase 2 clinical trial (NCT02649855) is evaluating the efficacy of PROSTVAC, either administered simultaneously with or subsequent to docetaxel, as a first-line treatment for patients with metastatic hormone-sensitive prostate cancer [179]. Another study (NCT04382898) is investigating the safety, tolerability, immunogenicity, and preliminary efficacy of the BNT112 cancer vaccine either as a monotherapy or in combination with cemiplimab in patients with mCRPC and high-risk localized prostate cancer [180]. Alongside these trials, NCT02325557 is focused on examining the utilization of ADXS31-142 alone or in combination with Pembrolizumab (MK-3475) in prostate cancer patients [181]. Furthermore, an additional study (NCT04989946), currently in the recruitment phase, aims to assess the utilization of ADT, either in conjunction with or without plasmid DNA vaccine encoding human androgen receptor (pTVG-AR), and with or without Nivolumab, for the treatment of patients who have been newly diagnosed with high-risk prostate cancer. These trials collectively signify an expansive and multifaceted research effort aimed at elucidating the potential benefits and applications of immunotherapy in the management of prostate cancer.

These trials represent essential efforts to explore the potential benefits and safety of combining immunotherapies with various treatment modalities in different stages of prostate cancer. The findings from these studies may contribute significantly to the development of more effective and personalized treatment approaches for prostate cancer patients. Key challenges include increased toxicity, optimal dosing and scheduling, predictive biomarkers to guide patient selection, and determining mechanisms of synergy or antagonism between combinations. Overall, rational combinations tailored to specific immune contexts may provide greater clinical benefit than single agents alone.

## 6. Challenges and Future Directions for Immunotherapy in Prostate Cancer

While immunotherapy has exhibited promising antitumor activity in subsets of patients with advanced prostate cancer, substantial challenges persist in elevating clinical efficacy and optimally integrating these approaches into the therapeutic paradigm.

A foremost obstacle is the relative paucity of neoantigens arising from somatic mutations, which restricts immunogenicity in prostate tumors compared to more responsive cancers [182,183]. The immunosuppressive ramifications of ADT may further hamper immune activation [54]. Moreover, redundancy across immunosuppressive mechanisms including checkpoint overexpression, modulatory cytokines, and suppressive cells contributes to immunotherapy resistance [184].

To surmount these barriers, an imperative future direction entails the development of rational combinatorial regimens that complement and synergize to collectively enhance immunogenicity and overcome immunosuppression. For example, coordinated immunotherapy platforms merging immune checkpoint inhibitors, CAR T cells, bispecific antibodies, or immunostimulatory cytokines with one another or conventional therapies may prove fruitful. However, escalated toxicity remains a concern with combination approaches, underscoring the parallel need for biomarkers to guide patient selection and optimize therapeutic indices.

Elucidating the precise mechanisms driving primary and acquired resistance also emerges as a research priority to inform second-generation therapeutic strategies. Molecular profiling of tumor and systemic immune parameters before and after progression on immunotherapies can illuminate aberrant pathways promoting immunosuppression or immune evasion. These insights may unveil actionable targets and support innovative approaches to reverse resistance.

Beyond combinations and elucidating resistance mechanisms, the identification of robust predictive biomarkers constitutes another pivotal step to enrich for potential immunotherapy responders. Harnessing technologies such as multiplex immunohistochemistry, liquid biopsy mutation tracking, and machine learning integration of multilayered datasets may enable the prospective identification of patients most likely to derive benefit [185,186]. Standardizing and clinically qualifying emergent biomarker signatures across large cohorts remains vital.

For cellular immunotherapies such as CAR T cells, optimization of construct designs, manufacturing, and host conditioning to enhance trafficking and maintain cytotoxic fitness in solid tumors warrants ongoing investigation. Moreover, earlier immunotherapy intervention, before the development of advanced castration-resistant disease, may prove fruitful by capitalizing on more intact immunity.

In summary, prostate cancer poses multifaceted challenges for immunotherapy but sustained progress in deciphering prostate cancer biology, unraveling resistance mechanisms, developing predictive biomarkers, and refining therapeutic strategies provides hope for realizing the full potential of immunotherapy to improve patient outcomes against this disease.

## 7. Conclusions

In conclusion, immunotherapy has recently emerged as an innovative treatment approach against advanced prostate cancer after decades of limited therapeutic progress. Strategies such as immune checkpoint inhibitors, bispecific antibodies, CAR T cell therapy, therapeutic vaccines, and cytokines aim to stimulate and harness endogenous anti-tumor immunity. Despite encouraging early clinical outcomes in subsets of patients, immunotherapy has not yet transformed survival outcomes in prostate cancer as seen in other tumor types.

Challenges such as low tumor mutation burden, immunosuppressive effects of androgen signaling, and redundancy across immunosuppressive mechanisms have constrained single-agent efficacy. However, rational combinatorial regimens, predictive biomarker development, deeper biological insights, and earlier intervention may help overcome these limitations and fulfill the potential of immunotherapy for prostate cancer. Ongoing phase 1–3 trials continue to evaluate novel immunotherapeutic agents and combinations in both metastatic castration-resistant and earlier stage prostate cancer settings. While early studies demonstrate immune activation and objective responses in some patients, definitive survival gains compared to standard of care have not yet materialized. Robust predictive biomarkers to prospectively identify patients likely to respond to different immunotherapies remain largely undefined.

Future research priorities include large biomarker-driven basket trials, unraveling mechanisms of primary and acquired resistance to guide next-generation designs, optimization of combination strategies matched to specific immune contexts, and elucidating optimal timing and sequencing with standard therapies. As the prostate cancer immunotherapy landscape rapidly evolves, it is hoped that these efforts will unlock its full potential and usher in a new era of significantly improved clinical outcomes. While questions remain, the accelerated progress provides hope for transforming the management of prostate cancer through immunotherapy in the future.

## Figures and Tables

**Figure 1 ijms-24-14347-f001:**
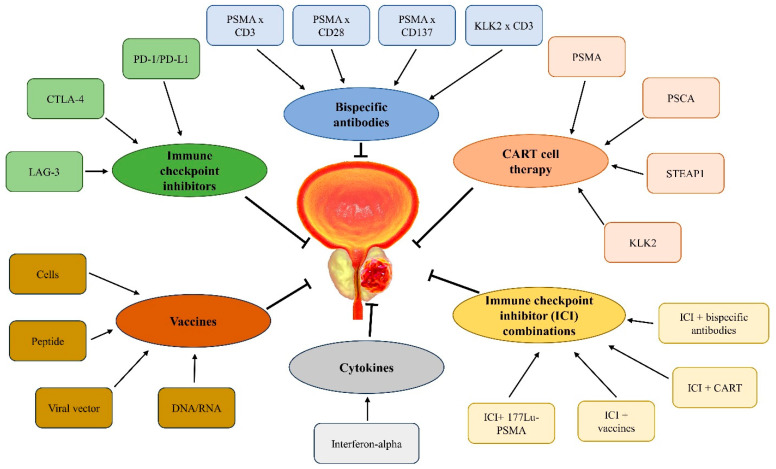
Overview of emerging immunotherapy approaches for prostate cancer.

**Table 2 ijms-24-14347-t002:** Key ongoing clinical trials testing novel immune checkpoint inhibitor combinations in metastatic prostate cancer.

NCT Number	Trial Name	Phase	Estimated Patients	Description	Sponsor
NCT03093428	N/A	2	45	Pembrolizumab + Radium-223 vs. Radium-223 in mCRPC	DFCI
NCT05766371	N/A	2	48	Pembrolizumab + 177Lu-PSMA-617 in mCRPC	UCSF
NCT03007732	N/A	2	23	Pembrolizumab +/− SD-101 in Hormone-Naïve Oligometastatic PCP with RT and iADT	UCSF
NCT01688492	N/A	1b/2	57	Ipilimumab + Abiraterone Acetate in Chemotherapy and Immunotherapy-naïve mCRPC	MSKCC
NCT02985957	CheckMate 650	2	351	Nivolumab + Ipilimumab, Ipilimumab Alone, or Cabazitaxel in mCRPC	Bristol-Myers Squibb
NCT03061539	N/A	2	380	Nivolumab Plus Ipilimumab followed by Nivolumab in mCRPC	UCL
NCT04446117	CONTACT-02	3	580	Atezolizumab + Carbozantinib vs. ARSI in mCRPC	Exelixis
NCT05150236	ANZUP2001	2	110	Nivolumab + Ipilimumab + 177 Lu-PSMA in mCRPC	ANUPCTG

Abbreviations: ANUPCTG: Australian and New Zealand Urogenital and Prostate Cancer Trials Group; ARSI: Androgen Receptor Signaling Inhibitor; DFCI: Dana-Farber Cancer Institute; iADT: Intermittent Androgen Deprivation Therapy; LLC: Limited Liability Company; Lu-PSMA: Lutetium PSMA; mCRPC: Metastatic Castration-Resistant Prostate Cancer; MSKCC: Memorial Sloan Kettering Cancer Center; N/A: not available: NCT: National Clinical Trial; NEPC: Neuroendocrine Prostate Cancer; PCP: Prostate Cancer Patients; RT: Radiotherapy; UCL: University College, London; UCSF: University of California, San Francisco.

**Table 3 ijms-24-14347-t003:** Representative ongoing BiTE trials in metastatic prostate cancer.

NCT Number	Phase	Estimated Patients	Description	Sponsor
NCT05369000	1/2	66	LAVA-1207 anti-PSMA × γδ	Lava Therapeutics
NCT04898634	1	160	JNJ-78278343 anti KLK2	Janssen Research & Development
NCT04740034	1	100	AMG 340 anti-PSMA × CD3	Amgen
NCT05125016	1/2	199	REGN4336/anti-PSMA × CD28 + Cemiplimab	Regeneron Pharmaceuticals
NCT04702737	1	41	AMG757/anti-DLL3 × CD3	Amgen
NCT04221542	1	464	AMG 509/anti-STEAP1 × CD3	Amgen
NCT04104607	1	86	anti-PSMA × CD3 CC-1	University Hospital Tuebingen

Abbreviations: DART: Dual Affinity Re-Targeting proteins; DLL3: Delta-like Protein 3; KLK2: Kallikrein 2; mCRPC: Metastatic Castration-Resistant Prostate Cancer; NCT: National Clinical Trial; PSMA: Prostate-Specific Membrane Antigen.

**Table 4 ijms-24-14347-t004:** Representative early phase CAR-T trials in advanced prostate cancer.

NCT Number	Phase	Estimated Patients	Description	Sponsor
NCT00664196	1	18	PSMA CAR-T + IL-2 in Advanced Prostate Cancer after Non-Myeloablative Conditioning (suspended)	Roger Williams Medical Center
NCT05732948	1	12	PD-1 Silent PSMA/PSCA Targeted CAR-T for Prostate Cancer	Shanghai Unicar-Therapy Bio-medicine Technology Co., Ltd.
NCT05805371	1	21	PSCA-Targeting CAR-T Plus or Minus Radiation in PSCA+ mCRPC	City of Hope Medical Center
NCT04249947	1	60	P-PSMA-101 CAR-T in mCRPC and Advanced Salivary Gland Cancers	Poseida Therapeutics, Inc.
NCT05022849	1	15	KLK2 CAR-T/JNJ-75229414 in mCRPC	Janssen Research & Development, LLC
NCT03089203	1	19	CART-PSMA-TGFβRDN Cells in mCRPC	University of Pennsylvania
NCT03873805	1	15	PSCA-CAR T Cells in Treating Patients with PSCA+ mCRPC	City of Hope Medical Center
NCT04227275	1	9	CART-PSMA-TGFβRDN in mCRPC	Tmunity Therapeutics
NCT04633148	1	39	UniCAR02-T Cells and PSMA Target Module in mCRPC with Progressive Disease after Standard Systemic Therapy	AvenCell Europe GmbH
NCT01140373	1	13	Adoptive Transfer of Autologous T Cells Targeted to PSMA in mCRPC	Memorial Sloan Kettering Cancer Center

Abbreviations: CAR-T: Chimeric Antigen Receptor T-cells; IL-2: Interleukin-2; mCRPC: Metastatic Castration-Resistant Prostate Cancer; NCT: National Clinical Trial; PD-1: Programmed Death-1; PSMA: Prostate-Specific Membrane Antigen; PSCA: Prostate Stem Cell Antigen; TGFβRDN: Transforming Growth Factor-beta Receptor Dominant Negative.

**Table 5 ijms-24-14347-t005:** Key vaccine trials in advanced prostate cancer.

NCT Number	Phase	Patients	Description	Sponsor
NCT00065442	3	512	Provenge^®^ (Sipuleucel-T) Active Cellular Immunotherapy in mCRPC	Dendreon
NCT03199872	1/2	22	RV001V, a RhoC Anticancer Vaccine, in Prostate Cancer	RhoVac APS
NCT01322490	3	1297	PROSTVAC-V/F +/− GM-CSF in mCRPC with Asymptomatic or Minimally Symptomatic Symptoms	Bavarian Nordic
NCT01341652	2	99	PAP Plus GM-CSF Versus GM-CSF Alone for Non-metastatic Prostate Cancer	University of Wisconsin, Madison
NCT01197625	1/2	30	DC-vaccination with Tumor mRNA in Curative Resected Prostate Cancer Patients	Oslo University Hospital
NCT05533203	1	24	Prodencel (an autologous dendritic cell therapeutic tumor vaccine) in mCRPC	Shanghai Humantech Biotechnology Co., Ltd.
NCT01436968	3	711	ProstAtak^®^ Immunotherapy With Standard Radiation Therapy for Localized Prostate Cancer	Candel Therapeutics, Inc.
NCT04701021	1	12	TENDU Vaccine in Prostate Cancer Patients with Relapse after Primary Radical Prostatectomy	Ultimovacs ASA
NCT00451022	3	750	Follow-Up Study of Subjects (including prostate cancer) Previously Enrolled in a Poxviral Vector Vaccine Trial	National Cancer Institute

Abbreviations: APC: Antigen Presenting Cells; DC: Dendritic Cell; GM-CSF: Granulocyte-Macrophage Colony-Stimulating Factor; mCRPC: Metastatic Castration-Resistant Prostate Cancer; PAP: Prostatic Acid Phosphatase.

**Table 6 ijms-24-14347-t006:** Key ongoing clinical trials of immunotherapeutic combinations in prostate cancer.

NCT Number	Phase	Estimated Patients	Description	Sponsor
NCT03532217	1	19	Neoantigen DNA Vaccine in Combination with Nivolumab/Ipilimumab and PROSTVAC in Hormone-Sensitive mCRPC	Washington University School of Medicine
NCT02649855	2	74	Docetaxel and PROSTVAC for mCRPC	NCI
NCT02325557	1/2	51	ADXS31-142 Alone and in Combination with Pembrolizumab in mCRPC	Advaxis, Inc
NCT04382898	1/2	115	PRO-MERIT in mCRPC	BioNTech SE
NCT04989946	1/2	39	ADT, +/− pTVG-AR, and +/− Nivolumab, in Newly Diagnosed, High-Risk Prostate cancer	University of Wisconsin, Madison
NCT04090528	2	60	pTVG-HP DNA Vaccine +/− pTVG-AR DNA Vaccine and Pembrolizumab in mCRPC	University of Wisconsin, Madison
NCT03600350	2	19	pTVG-HP and Nivolumab in Non-Metastatic PSA-Recurrent Prostate Cancer	University of Wisconsin, Madison
NCT03315871	2	29	Combination Immunotherapy in Biochemically Recurrent Prostate Cancer	NCI
NCT04114825	2	180	RV001V in Biochemical Failure Following Curatively Intended Therapy For Localized Prostate Cancer	RhoVac APS
NCT03493945	1/2	113	Immunotherapy Combination BN-Brachyury Vaccine, M7824, N-803 and Epacadostat in mCRPC	NCI
NCT02933255	1/2	29	PROSTVAC + Nivolumab in Prostate Cancer	NCI

Abbreviations: ADT: Androgen Deprivation Therapy; mCRPC: Metastatic Castration-Resistant Prostate Cancer; NCI: National Cancer Institute; NCT: National Clinical Trial; PSA: Prostate-Specific Antigen; pTVG-AR: plasmid DNA vaccine encoding human androgen receptor; pTVG-HP: plasmid DNA vaccine encoding human PAP.

## Data Availability

Not applicable.

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
