# Peer review of "Emerging Immunotherapy Approaches for Treating Prostate Cancer"

_ijms, 2023, doi:10.3390/ijms241814347_

Round 1

Reviewer 1 Report

This is an interesting, thorough, carefully and well-written work that provides a good overview of previous and ongoing research into prostate cancer immunotherapy, as well as challenges and future directions for improvement. Congratulations to the authors

I have only minor comments for improvement:

1-In the first part of the manuscript the authors could nuance a little more their statement that ICB does not work for CRPC by saying that, although most phase III trials showed no benefit of immunotherapy over standard therapy, responses could be observed, often in subgroups of patients, as was also the case in earlier phase I and II studies. In this respect, they could cite more studies reporting responses

In this support, in addition to Hansen keynote 028 and Keynote 199, it could be interesting to add

Graf RP et al. JAMA network open https://jamanetwork.com/journals/jamanetworkopen/fullarticle/2790572 In this comparative effectiveness study of 741 patients with mCRPC, patients with TMB of 10 mutations per megabase (mt/Mb) or greater had significantly longer time to next treatment and overall survival with ICIs vs taxanes.

Cabel et al J immuno cancer 2017 : https://pubmed.ncbi.nlm.nih.gov/28428880/ some complete responses with ipilimumab in CRPC

Graham L et al  Plos one 2020 . PSA50 responses to pembrolizumab occurred in 53% of cases menhttps://pubmed.ncbi.nlm.nih.gov/32453797/  

Also, I have not found mention to the report by Neeraj Agarwal et al. (combi IO + TKI) = phase I cosmic 021 atezo + cabo in mCRPC (https://pubmed.ncbi.nlm.nih.gov/35690072). Not a phase III, but potentially interesting to discuss

2- In the tables, it would be interesting for readers to have a reference number, if any, in a dedicated sub-column. I realize this may not be easy for formatting, but it may be worth a try.

3-L75-76: “prostate cancer had a considerably lower response rate than non-small cell lung cancer (NSCLC), melanoma, or renal cell carcinoma (RCC) to nivolumab. monotherapy (NCT00730639) [25, 26]“  could you be more specific and give numbers ?

4-For the benefit of readers, could you define the response rate criteria in this previous study and when relevant (PSA level? RESIST?)

Typos:

5-in table 1 : on placibo “Pembrolizumab + enzalutamide vs placibo +enzalutamide in mCRPC”

6-in figure 1 (PSMA x CD1437) do you mean CD137 ?

7-L74, typo in “…have been limited. in the Phase ½”

8-L79-80 KEYNOTE-199 trial: Authors could further discuss the subgroup analysis performed based on biomarkers (BRCA1 etc…)

9-L250: replace comma as appropriate in “BiTE therapies do carry certain pitfalls ,…”

10-L257: add a comma after of note “Of note high grade CRS is much less common with BiTEs compared to”

11-L376 : “o DLTs were observed, and 3 stable diseases by RECIST”

Author Response

 Point-to-Point Responses to Reviewer Comments

----------------------------------------------------------------------------

This is an interesting, thorough, carefully and well-written work that provides a good overview of previous and ongoing research into prostate cancer immunotherapy, as well as challenges and future directions for improvement. Congratulations to the authors

 I have only minor comments for improvement:

1-In the first part of the manuscript the authors could nuance a little more their statement that ICB does not work for CRPC by saying that, although most phase III trials showed no benefit of immunotherapy over standard therapy, responses could be observed, often in subgroups of patients, as was also the case in earlier phase I and II studies. In this respect, they could cite more studies reporting responses

In this support, in addition to Hansen keynote 028 and Keynote 199, it could be interesting to add

Graf RP et al. JAMA network open https://jamanetwork.com/journals/jamanetworkopen/fullarticle/2790572 In this comparative effectiveness study of 741 patients with mCRPC, patients with TMB of 10 mutations per megabase (mt/Mb) or greater had significantly longer time to next treatment and overall survival with ICIs vs taxanes.

Cabel et al J immuno cancer 2017 : https://pubmed.ncbi.nlm.nih.gov/28428880/ some complete responses with ipilimumab in CRPC

Graham L et al  Plos one 2020 . PSA50 responses to pembrolizumab occurred in 53% of cases menhttps://pubmed.ncbi.nlm.nih.gov/32453797/ 

Also, I have not found mention to the report by Neeraj Agarwal et al. (combi IO + TKI) = phase I cosmic 021 atezo + cabo in mCRPC (https://pubmed.ncbi.nlm.nih.gov/35690072). Not a phase III, but potentially interesting to discuss

Response: We really appreciate the reviewer’s suggestions and have included the following discussions into the section 2.2 in page 5 “While most phase III trials have not shown a definitive survival advantage, preliminary signs of immunotherapy activity have been observed in subsets of mCRPC patients, particularly in earlier phase studies. For instance, in a comparative effectiveness analysis of 741 mCRPC patients, ICIs improved outcomes versus taxanes for those with tumor mutational burden (TMB) 10 mutations/megabase, indicating TMB may predict ICI benefit [43]. Additionally, another study highlights that Ipilimumab can be effective in treating patients with CRPC, with some showing significant responses and no traceable residual disease [44]. In another study, pembrolizumab yielded PSA50 declines in 53% of MSI-high mCRPC cases, although the small sample size warrants further research  [45].  Furthermore, the COSMIC-021 phase 1b trial combining cabozantinib and atezolizumab showed encouraging activity, with an objective response rate of 23% in mCRPC. However, 55% experienced grade 3-4 adverse events, with pulmonary embolism, diarrhea, fatigue, and hypertension being the most prevalent [46].”

2- In the tables, it would be interesting for readers to have a reference number, if any, in a dedicated sub-column. I realize this may not be easy for formatting, but it may be worth a try.

Response: We appreciate this suggestion and have included references for each trial listed in Table 1. However, for the other tables summarizing ongoing trials, most of them lack published results, so we have not provided references.

3-L75-76: “prostate cancer had a considerably lower response rate than non-small cell lung cancer (NSCLC), melanoma, or renal cell carcinoma (RCC) to nivolumab monotherapy (NCT00730639) [25, 26]“  could you be more specific and give numbers ? 

Response: We appreciate this suggestion and have enhanced the sentence by incorporating additional information (page 4): “In the Phase 1/2 CheckMate 063 trial (NCT00730639), nivolumab monotherapy demonstrated varying response rates among different cancers. Specifically, non-small cell lung cancer (NSCLC), melanoma, and renal cell carcinoma (RCC) patients had objective response rates of 18%, 28%, and 27% respectively, as defined by RECIST criteria. However, there were no observed objective responses in those with colorectal or prostate cancer prostate cancer [6, 7].”

4-For the benefit of readers, could you define the response rate criteria in this previous study and when relevant (PSA level? RESIST?)

Response: We appreciate this suggestion and have included "as defined by RECIST criteria" in the revised description.

Typos:

5-in table 1 : on placibo “Pembrolizumab + enzalutamide vs placibo +enzalutamide in mCRPC”

6-in figure 1 (PSMA x CD1437) do you mean CD137 ?

7-L74, typo in “…have been limited. in the Phase ½”

8-L79-80 KEYNOTE-199 trial: Authors could further discuss the subgroup analysis performed based on biomarkers (BRCA1 etc…)

9-L250: replace comma as appropriate in “BiTE therapies do carry certain pitfalls ,…”

10-L257: add a comma after of note “Of note high grade CRS is much less common with BiTEs compared to”

11-L376 : “o DLTs were observed, and 3 stable diseases by RECIST”

Response: We really appreciate the reviewer's thorough review and recommendations. All typo errors have been addressed. Additionally, based on the reviewer's advice, we have expanded our discussion on the KEYNOTE-199 trial as follows (page 4): " Exploratory biomarker analysis identified higher response rates in subsets with DNA damage repair (DDR) aberrations, including 11% in patients with BRCA1/2 or ATM mutations [32]. This suggests genomic alterations enabling higher tumor immunogenicity like neoantigen burden and PD-L1 upregulation may enrich for ICI benefit".

Reviewer 2 Report

This is a really comprehensive and well-written review of high quality. I personally learned a lot by reading it. The authors' efforts in doing this review are well appreciated.

My only suggestion for this manuscript: It would make the manuscript easier to read (and make the review of immunotherapy easier to digest) by including a brief overview of the tumor immunological characteristics of prostate cancer at the beginning of the manuscript. It would be interesting to learn about some specific tumor microenvironment information about prostate cancer, before proceeding to the clinical strategies.

Author Response

 Point-to-Point Responses to Reviewer Comments

----------------------------------------------------------------------------

This is a really comprehensive and well-written review of high quality. I personally learned a lot by reading it. The authors' efforts in doing this review are well appreciated.

My only suggestion for this manuscript: It would make the manuscript easier to read (and make the review of immunotherapy easier to digest) by including a brief overview of the tumor immunological characteristics of prostate cancer at the beginning of the manuscript. It would be interesting to learn about some specific tumor microenvironment information about prostate cancer, before proceeding to the clinical strategies.

Response: We really appreciate this suggestion and have included the following paragraph in the introduction section (page 2) Unlike highly immunogenic tumors like melanoma, prostate cancer poses unique challenges for immunotherapy [6, 7]. The tumor microenvironment often displays immunosuppression, with enrichment of regulatory T cells (Tregs), myeloid-derived suppressor cells (MDSCs), and M2 macrophages, which can promote immune evasion [8]. Prostate tumors tend to harbor fewer neoantigens due to a relatively low tumor mutational burden, reducing their immunogenicity [9]. There is also marked upregulation of immunosuppressive molecules like CTLA-4 and DcR3 that attenuates anti-tumor immunity [10]. Furthermore, TGFβ is widely expressed in prostate cancer tumor cells and stromal cells and the expression of TGFβ is enriched in stromal cells of castration resistant prostate cancer and bone metastases [11-13]. Consequently, prostate cancers are often resistant to immunotherapies like checkpoint inhibitors that have shown efficacy in other malignancies. Comprehending these hurdles is vital for engineering more potent immunotherapies. Current research aims to gain deeper insights into the prostate cancer immunological landscape to create novel treatments that can overcome immunosuppression and trigger robust anti-tumor responses.